# Preparative Separation of Phenylethanoid and Secoiridoid Glycosides from Ligustri Lucidi Fructus by High-Speed Counter-Current Chromatography Coupled with Ultrahigh Pressure Extraction

**DOI:** 10.3390/molecules23123353

**Published:** 2018-12-18

**Authors:** Fengwei He, Li Chen, Qian Liu, Xiao Wang, Jia Li, Jinqian Yu

**Affiliations:** 1Liaoning Institute of Science and Technology, Benxi 117004, China; hfwdd@163.com (F.H.); chenlilycharming@126.com (L.C.); 2Shandong Key Laboratory of TCM Quality Control Technology, Shandong Analysis and Test Center, Qilu University of Technology (Shandong Academy of Sciences), Jinan 250014, China; 18765878227@126.com (Q.L.); wangx@sdas.org (X.W.); 3College of Pharmacy, Shandong University of Traditional Chinese Medicine, Jinan 250355, China

**Keywords:** high-speed counter-current chromatography, phenylethanoid glycoside, secoiridoid glycoside Ligustri Lucidi Fructus, ultrahigh pressure extraction

## Abstract

Three phenylethanoid glycosides, echinacoside (**1**), salidroside (**3**), and acteoside (**6**), and three secoiridoid glycosides, isonuezhenide (**2**), nuezhenoside G13 (**4**), and specnuezhenide (**5**), have been extracted and separated by a combined method of ultrahigh pressure extraction (UPE) and high-speed counter-current chromatography (HSCCC) from Ligustri Lucidi Fructus. For the UPE, the optimal extraction was developed with conditions including solvent of 90% ethanol, sample to solvent ratio of 1:20 g/mL, pressure of 200 MPa, and time of 2 min, which rendered the yields of compounds **4** and **5** were 15.0 and 78.0 mg/g, respectively. For the HSCCC separation, the strategy of changing flow rates between 1.0 and 2.0 mL/min allowed the acquisition for 2.7 mg of compound **1**, 4.5 mg of compound **2**, 6.8 mg of compound **3**, 5.9 mg of compound **4**, 11.2 mg of compound **5**, and 2.2 mg of compound **6** in one separation run under the solvent system of ethyl acetate:*n*-butanol:water (2:1:3, *v*/*v*) from 200 mg of the UPE extract. The structures of these phenylethanoid and secoiridoid glycosides were elucidated by extensive spectroscopic methods.

## 1. Introduction

*Ligustrum lucidum Ait.*, an evergreen shrub from the family of Oleaceae, is mainly distributed in the southern regions of China, such as Jiangsu, Guangxi, Yunnan and so on. The dried mature fruits of *L*. *lucidum*, documented in the Chinese Pharmacopoeia as “Ligustri Lucidi Fructus”, have long been used in traditional Chinese medicine to invigorate muscles and bones, and tonify kidney and liver [1]. Additionally, Ligustri Lucidi Fructus was also recorded as a tonic and dietary supplement possessing the homology of medicine and food [2,3]. Chemical investigations of the fruit have led to the isolation and identification of its main native organic compounds as triterpenoids, phenylethanoid glycosides, and secoiridoid glycosides [4,5,6,7], with the polar of the latter two kinds being higher and having been revealed to possess anti-aging, anti-inflammatory, antioxidant, immunomodulatory, antidiabetic, and antitumor activities [8,9]. A literature survey indicated that phenylethanoid glycosides in Ligustri Lucidi Fructus were mainly composed of salidroside, echinacoside, acteoside, and cimidahurinine, as well as secoiridoid glycosides mainly composed of specnuezhenide, isonuezhenide, and nuezhenoside G13 [7]. Additionally, salidroside and specnuezhenide are representative compounds of Ligustri Lucidi Fructus, with the latter used as the quality control marker in the Chinese Pharmacopoeia 2015 [10]. However, the extraction and separation of these polar compounds by some conventional techniques, such as ultrasonic and heating extraction and column chromatography separation, often leads to high consumption of solvent, high time consumption, and irreversible adsorption of the polar compounds [11,12].

In terms of employing an effective extraction and separation method for the phenylethanoid and secoiridoid glycosides, the application of ultrahigh pressure extraction (UPE) and high-speed counter-current chromatography (HSCCC) could be a better choice. This UPE pretreatment technique is widely used due to its advantageous non-thermal nature and time saving potential, which can cause some cell structural changes in plants which enhance the velocity of solvent permeability, increasing not only the mass transfer rate but also the active component diffusion when pressure is increased between 100 and 800 MPa, rendering the ingredients more available to be extracted [13]. HSCCC is a liquid–liquid partition chromatography, which has been gaining popularity not only because of the partition of compounds between two immiscible solvent phases (stationary and mobile phases) but also due to its none irreversible adsorption of compounds [14], and consequently, preservation of the intact components of the plant.

In this study, a combined method for extraction and separation of three phenylethanoid glycosides, echinacoside (**1**), salidroside (**3**), and acteoside (**6**), and three secoiridoid glycosides, isonuezhenide (**2**), nuezhenoside G13 (**4**), and specnuezhenide (**5**) (Figure 1), from *L*. *lucidum*, has been successfully conducted by UPE and HSCCC.

## 2. Results

### 2.1. Ultrahigh Pressure Extraction Parameters

The UPE extraction methodology was developed after measuring the individual effects of four major parameters using single-factor experiments. The effects of ethanol concentration (50%, 70%, 90%, 100%), sample to solvent ratio (1:10, 1:20, 1:30), pressure (100, 200, 300 MPa), and time (1, 2, 3 min) on extraction efficiency of the main phenylethanoid and secoiridoid glycosides were compared to obtain the optimal extraction conditions. Two variable values, the concentrations of nuezhenoside G13 (**4**) and specnuezhenide (**5**), were selected as the characteristic ingredients for extraction efficiency, because of their relative higher amounts.

The selection of a suitable solvent could strongly affect the extraction efficiency of the medicine, which depends on its affinity with the target components (i.e. their similar polarities). According to the polarity of the targeted phenylethanoid and secoiridoid glycosides in Ligustri Lucidi Fructus, ethanol with different concentrations was selected as the extracted solvent for UPE. Herein, four different ethanol concentrations (50%, 70%, 90%, and 100%) on the extracted amounts of compounds **4** and **5** were evaluated for UPE, with the three other parameters as 1:10 g/mL of sample to solvent ratio, 1 min, and 100 MPa. Figure 2a indicated that the yields of **4** and **5** extracted from Ligustri Lucidi Fructus increased when the ethanol concentration ascended from 50% to 90%, with yields slightly increased from 50% to 70% (percentages of content change for **4** and **5** was 15.7% and 11.8%, respectively), greatly increased from 70% to 90% (percentages of content change for **4** and **5** as 74.7% and 51.2%, respectively), but slightly decreased from 90% to 100% (percentages of content change for **4** and **5** as −3.2% and −1.7%, respectively). Therefore, 90% of ethanol was selected as the suitable extraction solvent for UPE.

To determine the effect of the second parameter, three different sample to solvent ratios (1:10, 1:20, 1:30 g/mL) on the extracted amounts of compounds **4** and **5** were investigated for UPE, with the three other parameters as 90% of ethanol, 100 MPa, and 1 min. We found that the contents of the two target compounds increased along with smaller tested sample to solvent ratios as shown in Figure 2b. When decreasing the sample to solvent ratios from 1:10 g/mL to 1:20 g/mL, the yields of compounds **4** and **5** rose noticeably with percentages of content change as 74.7% and 51.2%, respectively. When the sample to solvent ratios were changed from 1:20 g/mL to 1:30 g/mL, however, the extraction yields of **4** and **5** rose linearly. In general, increased solvent amount could lead to more ingredients dissolved out. When the ingredients were almost dissolved out, the extraction efficiency would not increase noticeably, although the sample to solvent ratios were decreased. According to the above factors, the suitable sample to solvent ratio was fixed at 1:20 g/mL.

To determine the effect of the third parameter, three different pressures (100, 200, 300 MPa) on the extracted contents of compounds **4** and **5** were investigated for UPE, with the three other parameters as 90% of ethanol, sample to solvent ratio of 1:20 g/mL, and 1 min. Figure 2c showed that the extraction yields of compounds **4** and **5** displayed the same trends as those in Figure 2b when increasing pressure from 100 MPa to 300 MPa, where the variation trends of the extraction yields from 100 MPa to 200 MPa and from 200 MPa to 300 MPa firstly rose noticeably and then rose linearly. These results were possibly due to ultrahigh pressure enhancing the speed of solvent permeability and solute transfer, thereby leading to a higher extraction efficiency of constituents with higher pressures. Considering the results, UPE extraction at 200 MPa was determined as the optimal condition.

For the fourth parameter, solvent of 90% ethanol, sample to solvent ratio of 1:20 g/mL, and pressure of 200 MPa were used to extract phenylethanoid and secoiridoid glycosides from Ligustri Lucidi Fructus with different extraction times (1, 2, 3 min), and the yield results are shown in Figure 2d. Under this condition, the contents of both compounds increased noticeably with the time varying from 1 min to 2 min, but the trend rose linearly with the time varying from 2 min to 3 min. Prolonged extraction time may lengthen the contact time for solvent and solute, and then produce a higher yield. When equilibrium of solute distributing in and out of the cells was obtained, the yield would not increase significantly, although the holding time was prolonged. Thus, the optimal extraction time was regarded as 2 min.

Finally, the optimized UPE pretreatment methodology was considered as solvent of 90% ethanol, sample to solvent ratio of 1:20 g/mL, pressure of 200 MPa, and time of 2 min, which were a result from the combined tests above. As per the UPE pretreatment conditions, the extraction yields of quantified compounds **4** and **5** from Ligustri Lucidi Fructus were 15.0 and 78.0 mg/g, respectively.

The extractabilities of phenylethanoid and secoiridoid glycosides from Ligustri Lucidi Fructus carried out by UPE and traditional heat reflux and ultrasonic experiments were compared. The traditional extractions were carried out with the same extraction time of 1 h and sample to solvent ratio of 1:20 g/mL. The analysis results of the crude extract obtained by HPLC chromatograms of the three different treatment methods were similarly characterized, except for more extraction yields of compounds **3**, **4**, and **5** by UPE than the other two extractions in Figure 3. Moreover, to achieve the same amounts of purified compounds **3**, **4**, and **5****,** the extraction time and solvent amount of the two traditional extraction methods were much more than those of UPE. Thus, the UPE pretreatment was confirmed to be efficient for extraction of phenylethanoid and secoiridoid glycosides from Ligustri Lucidi Fructus.

### 2.2. Optimization of the High-Speed Counter-Current Chromatography Conditions

Six compounds, including three phenylethanoid glycosides, echinacoside (**1**), salidroside (**3**), and acteoside (**6**), and three secoiridoid glycosides, isonuezhenide (**2**), nuezhenoside G13 (**4**), and specnuezhenide (**5**), from Ligustri Lucidi Fructus ranged broadly from 14 min to 42 min in the HPLC chromatogram. Thus, to select a robust solvent system was the first point of all in a HSCCC separation, in which the expected dissociation (*K*_D_) values for the six compounds are usually between 0.5 and 2.0 [13,14,15]. The solvent system composed of ethyl acetate:*n*-butanol:water was selected as the experimental solvent, according to the polarities and separating experiences of such compounds [16]. In this study, three different solvent systems with solvent ratios of ethyl acetate:*n*-butanol:water as 1:0:1, 4:1:5, and 2:1:3 were used to run HSCCC separations, and the *K*_D_ values for the six compounds were calculated in Table 1. Fortunately, the appropriate *K*_D_ values could be achieved when the volume ratios of ethyl acetate:*n*-butanol:water were 4:1:5 and 2:1:3. When the solvent system of ethyl acetate:*n*-butanol:water (4:1:5, *v/v*) was tested, the first three compounds (**1**, **2**, **3**) with relatively higher polarity were speedily eluted out as two peaks by HSCCC, with compounds **1** and **2** mixed in the first peak and compounds **2** and **3** mixed in the second peak, as shown in Figure 4a. Next, the other solvent system of ethyl acetate:*n*-butanol:water (2:1:3, *v/v*) was tested, which rendered the first three compounds (**1**, **2**, **3**) separated from each other in three tightly coupled peaks by HSCCC, with a flow rate of 2.0 mL/min, shown in Figure 4b. To achieve complete separation for the first three compounds, the flow rate was slowed down to 1.0 mL/min, shown in Figure 4c. Compared with Figure 4b, the first three compounds were separated from each other very well, however, the separation time doubled. To save time for HSCCC, a lower flow rate of 1.0 mL/min was used in the time range from 100 min to 170 min, when compounds **1**, **2**, and **3** emerged, and then the higher flow rate of 2.0 mL/min was used in the last 130 min, as shown in Figure 4d. Finally, the separation strategy was determined to combine two different flow rates (1.0 and 2.0 mL/min) successively in one HSCCC run, which improved not only the peak resolutions for the first three compounds, but also the separation efficiency.

Besides the two established factors, the solvent system, and the flow rate, the revolution speed of the separation column was also investigated. Three experiments changing the revolution speed (700, 800, and 850 rpm) were carried out, which resulted in the revolution speed of 800 rpm being optimal. 

### 2.3. Purification of Phenylethanoid and Secoiridoid Glycosides by HSCCC 

200 mg of Ligustri Lucidi Fructus UPE extract was successfully separated by HSCCC using two changeable flow rates by the solvent system of ethyl acetate:*n*-butanol:water (2:1:3, *v*/*v*). As shown in Figure 4d, the separation was initiated with a flow rate at 2.0 mL/min, and once compound **1** emerged, the flow rate was adjusted to 1.0 mL/min. After compound **3** was completely eluted out, the flow rate was increased to 2.0 mL/min again. Subsequently, the surplus three compounds were obtained in the next 130 min. Finally, after the single HSCCC run, 2.7 mg of echinacoside (**1**), 4.5 mg of isonuezhenide (**2**), 6.8 mg of salidroside (**3**), 5.9 mg of nuezhenoside G13 (**4**), 11.2 mg of specnuezhenide (**5**), and 2.2 mg of acteoside (**6**) were obtained from 200 mg of the UPE extract, with purities all over 90.0% determined by HPLC.

### 2.4. Structure Identification of the Isolated Compounds

The structure elucidations of compounds **1**–**6** were finally achieved by analyzing the high resolution electrospray ionization mass spectra (HRESI-MS) and nuclear magnetic resonance (NMR) spectroscopic data and comparing with data from the literature [17,18,19,20]. Finally, compounds **1**–**6** were identified as echinacoside (**1**), isonuezhenide (**2**), salidroside (**3**), nuezhenoside G13 (**4**), specnuezhenide (**5**), and acteoside (**6**), respectively.

*Echinacoside* (**1**): HRESI-MS *m*/*z* [M + Na]^+^: 809.1837; ^1^H-NMR (CD_3_OD, 400 MHz) δ: 7.47 (1H, d, *J* = 16.0 Hz, H-7′′′′), 7.03 (1H, brs, H-2′′′′), 6.99 (1H, brd, *J* = 8.4 Hz, H-6′′′′), 6.76 (1H, d, *J* = 8.4 Hz, H-5′′′′), 6.64 (1H, brs, H-2), 6.62(1H, d, *J* = 8.0 Hz, H-5), 6.51 (1H, dd, *J* = 1.6, 8.0 Hz, H-6), 6.21 (1H, d, *J* = 16.0 Hz, H-8′′′′), 5.03 (1H, brs, H-1′′), 4.71 (1H, t, *J* = 9.6 Hz, H-4′), 4.37 (1H, d, *J* = 8.0 Hz, H-1′), 4.16 (1H, d, *J* = 7.6 Hz, H-1′′′), 3.90 (1H, m, H-α), 2.70 (2H, m, H_2_-β), 0.95 (1H, d, *J* = 6.0 Hz, H-6′′); ^13^C-NMR (CD_3_OD, 100 MHz) δ: 166.5 (C-9′′′′), 149.2 (C-4′′′′), 146.2 (C-7′′′′), 145.8 (C-3′′′′), 145.4 (C-3), 144.0 (C-4), 129.4 (C-1), 125.8 (C-1′′′′), 122.0 (C-6′′′′), 120.0 (C-6), 116.8 (C-2), 116.2 (C-5′′′′), 115.9 (C-5), 115.3 (C-2′′′′), 113.6 (C-8′′′′), 103.8 (C-1′′′), 102.5 (C-1′), 101.6 (C-1′′), 79.2 (C-3′), 77.3 (C-5′′′), 77.0 (C-3′′′), 74.8 (C-2′), 73.9 (C-2′′′), 73.6 (C-5′), 72.1 (C-4′′), 71.0 (C-8), 70.8 (C-3′′), 70.7 (C-2′′), 70.4 (C-4′′′), 69.5(C-4′), 69.1 (C-5′′), 68.5 (C-6′), 61.5 (C-6′′′), 35.5 (C-7), 18.6 (C-6′′).

*Isonuezhenide* (**2**): HRESI-MS *m*/*z* [M + Na]^+^: 709.1438; ^1^H-NMR (CD_3_OD, 400 MHz) δ: 7.52(1H, brs, H-3), 7.30 (2H, d, *J* = 8.4 Hz, H-2′′′, 6′′′), 6.99 (2H, d, *J* = 8.4 Hz, H-3′′′, 5′′′), 6.18 (1H, q, *J* = 6.8 Hz, H-8), 5.97 (1H, brs, H-1), 4.81 (1H, overlapped, H-1′), 4.30 (1H, d, *J* = 8.0 Hz, H-1′′), 4.11 (1H, m, H-α), 3.81 (1H, m, H-α), 2.94 (2H, t, *J* = 7.6 Hz, H_2_-β), 2.74 (1H, m, Ha-6), 2.51 (1H, m, Hb-6), 1.75 (3H, dd, *J* = 1.0, 7.0 Hz, H-10); ^13^C-NMR (CD_3_OD, 100 MHz) δ: 170.4 (C-7), 167.2 (C-11), 153.4 (C-3), 148.5 (C-4′′′), 134.9 (C-1′′′), 129.6 (C-2′′′, 6′′′), 129.5 (C-9), 123.5 (C-8), 121.1 (C-3′′′, 5′′′), 107.9 (C-4), 103.0 (C-1′′), 99.3 (C-1′), 93.5 (C-1), 77.0 (C-5′), 76.6 (C-3′), 76.5 (C-5′′), 73.7 (C-3′′), 73.4 (C-2′), 70.3 (C-4′′), 70.08 (C-4′, α), 70.03 (C-2′′), 61.39 (C-6′), 61.25 (C-6′′), 50.5 (OMe), 40.0 (C-6), 33.9 (C-β), 30.3 (C-5), 12.3 (C-10).

*Salidroside* (**3**): HRESI-MS *m*/*z* [2M + Na]^+^: 623.2222; ^1^H-NMR (CD_3_OD, 400 MHz) δ: 7.06 (2H, d, *J* = 8.4 Hz, H-2, 6), 6.69 (2H, d, *J* = 8.4 Hz, H-3, 5), 4.29 (1H, d, *J* = 8.0 Hz, H-1′), 4.02 (1H, m, H-α), 3.86 (1H, dd, *J* = 1.6, 12.4 Hz, Ha-6′), 3.68 (2H, overlapped, H-α, Hb-6′), 3.23-3.37 (3H, overlapped, H-3′, 4′, 5′), 3.18(1H, t, *J* = 8.8 Hz, H-2′), 2.83 (2H, t, *J* = 8.0 Hz, H_2_-β); ^13^C-NMR (CD_3_OD, 100 MHz) δ: 155.4 (C-1), 129.6 (C-3,5), 129.4 (C-4), 114.7 (C-2,6), 103.0 (C-1′), 76.7 (C-4′), 76.6 (C-3′), 73.7 (C-2′), 70.7 (C-5′), 70.3 (C-8), 61.4 (C-6′), 35.0 (C-7).

*Nuezhenoside G13* (**4**): HRESI-MS *m*/*z* [2M + Na]^+^: 1095.4089; ^1^H-NMR (CD_3_OD, 400 MHz) δ: 7.57 (1H, brs, H-3A), 7.52 (1H, brs, H-3B), 7.29 (2H, d, *J* = 8.4 Hz, H-2′′′, 6′′′), 6.98 (2H, d, *J* = 8.4 Hz, H-3′′′, 5′′′), 6.18 (1H, q, *J* = 6.8 Hz, H-8A), 6.09 (1H, q, *J* = 6.8 Hz, H-8B), 6.03 (1H, brs, H-1A), 5.92 (1H, brs, H-1B), 4.81 (2H, d, *J* = 7.6 Hz, H-1′A,1′B), 4.31 (1H, d, *J* = 8.0 Hz, H-1′′), 4.01 (1H, m, H-α), 3.76 (1H, m, H-α), 3.73 (3H, s, OMe-A), 3.68 (3H, s, OMe-B), 2.94 (2H, t, *J* = 7.6 Hz, H_2_-β), 1.76 (1H, d, *J* = 6.8 Hz, H-10A), 1.72 (1H, d, *J* = 6.8 Hz, H-10B); ^13^C-NMR (CD_3_OD, 100 MHz) δ: 171.6 (C-7A), 170.3 (C-7B), 167.3 (C-11A, 11B), 153.9 (C-3A), 153.8 (C-3B), 149.1 (C-4′′′), 136.6 (C-1′′′), 129.6 (C-2′′′, 6′′′), 129.2 (C-9A), 129.1 (C-9B), 123.8 (C-8A), 123.6 (C-8B), 121.1 (C-3′′′, 5′′′), 108.0 (C-4A), 107.9 (C-4B), 103.0 (C-1′′), 99.6 (C-1′A), 99.5 (C-1′B), 94.0 (C-1A), 93.8 (C-1B), 77.0 (C-3′A, 3′B), 76.5 (C-5′A, 5′B), 76.5 (C-3′′), 73.8 (C-5′′), 73.6 (C-2′′), 73.4 (C-2′A, 2′B), 70.2 (C-4′A, 4′B), 70.1(C-α), 70.0 (C-4′′), 63.6 (C-6′′), 61.34 (C-6′A), 61.27 (C-6′B), 50.6 (OMe-A, B), 39.9 (C-6A), 39.7 (C-6B), 35.2 (C-β), 30.4 (C-5A,5B), 12.4 (C-10A), 12.3 (C-10B).

*Specnuezhenide* (**5**): HRESI-MS *m*/*z* [M + Na]^+^: 709.2308; ^1^H-NMR (CD_3_OD, 400 MHz) δ: 7.52 (1H, brs, H-3), 7.05 (2H, d, *J* = 8.4 Hz, H-2′′′, 6′′′), 6.63 (2H, d, *J* = 8.4 Hz, H-3′′′, 5′′′), 6.09 (1H, q, *J* = 7.2 Hz, H-8), 5.92 (1H, brs, H-1), 4.81 (1H, d, *J* = 7.6 Hz, H-1′), 4.30 (1H, d, *J* = 8.0 Hz, H-1′′), 3.68 (3H, s, OMe), 2.83 (2H, t, *J* = 9.2 Hz, H_2_-β), 2.74 (1H, dd, *J* = 4.8, 14.4 Hz, Ha-6), 2.49 (1H, dd, *J* = 8.8, 14.4 Hz, Hb-6), 1.72 (3H, d, *J* = 7.2 Hz, H-10); ^13^C-NMR (CD_3_OD, 100 MHz) δ: 171.6 (C-7), 167.3 (C-11), 155.5 (C-4′′′), 153.8 (C-3), 129.6 (C-3′′′, 5′′′), 129.3 (C-4), 129.1 (C-1′′′), 123.6 (C-8), 114.8 (C-2′′′, 6′′′), 108.0 (C-9), 103.0 (C-1′′), 99.5 (C-1′), 93.8 (C-1), 77.0 (C-3′′), 76.6 (C-5′′), 76.5 (C-5′), 73.8 (C-3′), 73.6 (C-2′), 73.4 (C-2′′), 70.8 (C-4′′), 70.2 (C-α), 70.1 (C-4′), 63.6 (C-6′′), 61.3 (C-6′), 50.6 (OMe), 39.9 (C-6), 35.0 (C-β), 30.4 (C-5), 12.3 (C-10).

*Acteoside* (**6**): HRESI-MS *m*/*z* [M + Na]^+^: 647.1963; ^1^H-NMR (CD_3_OD, 400 MHz) δ: 7.59 (1H, d, *J* = 15.6 Hz, H-7′′′), 7.05 (1H, d, *J* = 1.6 Hz, H-2′′′), 6.95 (1H, dd, *J* = 2.0, 8.4 Hz, H-6′′′), 6.77 (1H, d, *J* = 8.0 Hz, H-5′′′), 6.69 (1H, d, *J* = 2.0 Hz, H-2), 6.67 (1H, d, *J* = 8.0 Hz, H-5), 6.56 (1H, dd, *J* = 2.0, 8.0 Hz, H-6), 6.27 (1H, d, *J* = 15.6 Hz, H-8′′′), 5.18 (1H, brs, H-1′′), 4.37 (1H, d, *J* = 8.0 Hz, H-1′), 4.04 (1H, m, H-α), 2.79 (2H, m, H_2_-β), 1.09 (1H, d, *J* = 6.0 Hz, H-6); ^13^C-NMR (CD_3_OD, 100 MHz) δ: 166.9 (C-9′′′), 148.4 (C-4′′′), 146.6 (C-7′′′), 145.4 (C-3′′′), 144.7 (C-3), 143.2 (C-4), 130.1 (C-1), 126.1 (C-1′′′), 121.8 (C-6′′′), 119.9 (C-6), 115.7 (C-2), 115.1 (C-5), 114.9 (C-5′′′), 113.8 (C-2′′′), 113.3 (C-8′′′), 102.8 (C-1′), 101.6 (C-1′′), 80.2 (C-3′), 74.8 (C-5′), 74.7 (C-2′), 72.4 (C-4′′), 71.0 (C-3′′), 70.9 (C-8), 70.7 (C-2′′), 69.2 (C-5′′), 69.0 (C-4′), 61.0 (C-6′), 35.2 (C-7), 17.0 (C-6′′).

## 3. Materials and Methods

### 3.1. Reagents and Materials

Ligustri Lucidi Fructus, the dried mature fruits of *L*. *lucidum*, was purchased from Zhonglu hospital of Shandong University of Traditional Chinese Medicine and authenticated by Prof. Jia Li (Shandong University of Traditional Chinese Medicine). All solvents used, except for acetonitrile, which was of HPLC grade and used in HPLC analysis, were of analytical grade. Acetonitrile was purchased from Fisher Scientific (Tedia Company, Fairfield, CT, USA), and other solvents were purchased from Fuyu Fine Chemical Co., Ltd (Tianjin, China). The water used was deionized by an osmosis Milli-Q system (Millipore, Bedford, MA, USA).

### 3.2. Apparatus 

The extraction was carried out by the HPP.L3-600 UPE with a High Hydrostatic Pressure Processor (Huataisenmiao Biology Engineering Technology Co. Ltd., Tianjin, China). The separations were performed on a TBE-300A high-speed counter-current chromatography equipment (Tauto Biotechnique, Shanghai, China), fitted with a polytetrafluoroethylene (PTFE) multilayer coil 300 mL capacity, 1.6 mm in diameter, and a 20 mL manual sample loop. The revolution speed of the column coil was regulated to be 800 rpm. The HSCCC system was equipped with a Model 3057 portable recorder (Yokogawa, Sichuan Instrument Factory, Sichuan, China), a 8823A-UV detector at 254 nm (Beijing Emilion Technology, Beijing, China), a TBP-5002 constant-flow pump (Tauto Biotechnique, Shanghai, China), as well as a DC-0506 low constant temperature-circulating bath (Tauto Biotechnique, Shanghai, China) to maintain the temperature at 25 °C.

Waters e2695 equipment was used to perform HPLC analysis. The Waters 2695 system consisted of an Empower 3 ChemStation unit, a Waters 2695 solvent delivery unit, a Waters 2998 Photodiode Array Detection (DAD) detector, an autosampler, and a Waters 2695 column oven. The analytical column used was a PurospherSTAR RP_18_Endiapped (250 mm × 4.6 mm, 5 μm). The NMR data of the isolated compounds were collected using a Bruker AV-400 spectrometer (Bruker BioSpin, Rheinstetten, Germany) and HRESI-MS analysis was performed on a Bruker Impact II mass spectrometer (Bruker Daltonic Inc., Bremen, Germany).

### 3.3. Ultrahigh Pressure Extraction 

The extraction was carried out in a sterile polyethylene bag containing 1.0 g of Ligustri Lucidi Fructus powder and 20 mL of 90% ethanol. After being sealed up using a plastic-envelop machine, the packaged bag was subjected to a pressure vessel from the HPP.L 3-600 equipment, and then extracted for a certain period. After extraction, the extracts were removed from the bag and filtered under reduced pressure, and then the filtrate was concentrated to yield the initial UPE extract. After the UPE extraction factors were determined, 2.0 g Ligustri Lucidi Fructus powder was extracted under the determined UPE conditions, and 0.341 g of 90% ethanol crude extract was obtained, which was stored at 2–8 °C for the subsequent HSCCC separation.

### 3.4. Traditional Extraction

The procedure for heat extraction was as follows: The dried and powdered fruits (1.0 g) were extracted in a 50 mL flask by 90% ethanol (20 mL) at 60 °C for 1 h. 

The procedure for ultrasonic extraction was as follows: The dried and powdered fruits (1.0 g) were decocted in a 50 mL flask by 90% ethanol (20 mL) at room temperature for 1 h.

### 3.5. HSCCC Separation Procedure 

For the HSCCC experiment, a TBE-300A HSCCC apparatus was used, along with the selected solvent system of ethyl acetate:*n*-butanol:water (2:1:3, *v*/*v*). Firstly, the solvent mixture was equilibrated in a separator funnel, resulting in two phases, with the upper one as the stationary phase and the lower one as the mobile phase. Next, the HSCCC separation began. The column was entirely filled with the upper phase at 20.0 mL/min, which was subsequently revolved at a speed of 800 rpm, while the lower phase was pumped into the column at 2.0 mL/min. When the lower phase was eluted out, the steady equilibrium state of the solvent system was reached; the sample solvent (each 5 mL for upper and lower phase) containing 200 mg of extract was injected into the apparatus through the sample loop, and the lower phase was pumped into the column with a changeable flow rate: 0–100 min, 2.0 mL/min; 100–170 min, 1.0 mL/min; 170–300 min, 2.0 mL/min. Separation curves were monitored by UV at 254 nm, and the effluents were collected together every 10 mL. Finally, when the HSCCC separation was over, ethanol was used as the mobile phase to completely elute the residual solvents to calculate the stationary phase retention.

### 3.6. HPLC Analysis and Identification of the Fractions

HPLC analysis of the crude sample by UPE and fractions obtained from the HSCCC separation were performed on a Waters e2695 apparatus. Acetonitrile (A)-water (B) was appointed as the mobile phase, and the gradient elution was: 0–2 min, 5–5% A; 2–40 min, 5–30% A; 40–45 min, 30–30% A; 45–53 min, 30–50% A, at a flow rate of 1.0 mL/min. The HPLC chromatograms were carried out by a Purospher STAR RP18 Endiapped (250 mm × 4.6 mm i.d., 5 μm) and detected at 282 nm. The structure elucidations of these compounds were achieved by analyzing the HRESI-MS and NMR spectroscopic data and comparing with data in the literature. 

## 4. Conclusions

In this present work, a combined method for rapid extraction and efficient separation of three phenylethanoid glycosides, echinacoside (**1**), salidroside (**3**), and acteoside (**6**), and three secoiridoid glycosides, isonuezhenide (**2**), nuezhenoside G13 (**4**), and specnuezhenide (**5**), from Ligustri Lucidi Fructus, has been successfully achieved by UPE and HSCCC for the first time. The optimal UPE extraction was developed with conditions set at solvent of 90% ethanol, sample to solvent ratio of 1:20 g/mL, pressure of 200 MPa, and time of 2 min, which rendered the yields of **4** and **5** to be 15.0 and 78.0 mg/g, respectively. These experimental results indicated that UPE extraction needed shorter time and less solvent, however, produced higher yields when compared with conventional extraction methods. Meanwhile, the HSCCC separation strategy of changing flow rates between 1.0 and 2.0 mL/min allowed the acquisition of these six compounds in one separation run in about 300 min, which indicated that HSCCC could yield compounds of widely varied polarities with less time and solvent consumption compared with conventional column separation. The overall results showed that the combination of UPE and HSCCC is efficient for the extraction and isolation of phenylethanoid and secoiridoid glycosides from Ligustri Lucidi Fructus. To the best of our knowledge, this integration method of UPE and HSCCC was applied rapidly to obtain the active components from Ligustri Lucidi Fructus for the first time. Furthermore, this gradient flow rate method demonstrated good performance in isolating compounds with similar polarities and poor peak resolutions, which paved the pathway for one-run systematic separation of complex compounds.

## Figures and Tables

**Figure 1 molecules-23-03353-f001:**
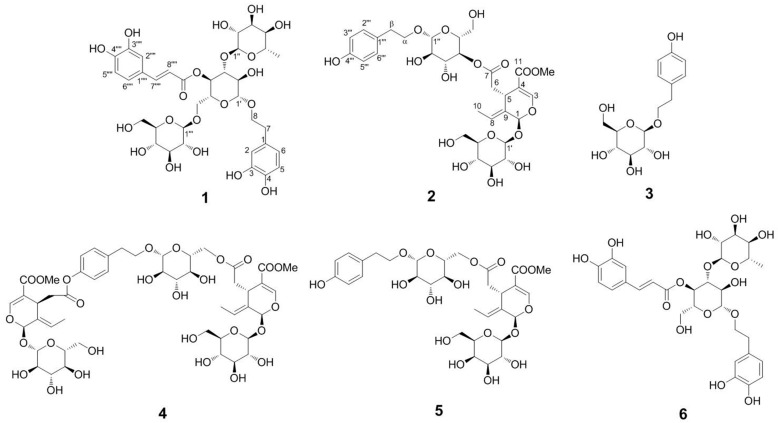
Chemical structures of compounds **1**–**6**.

**Figure 2 molecules-23-03353-f002:**
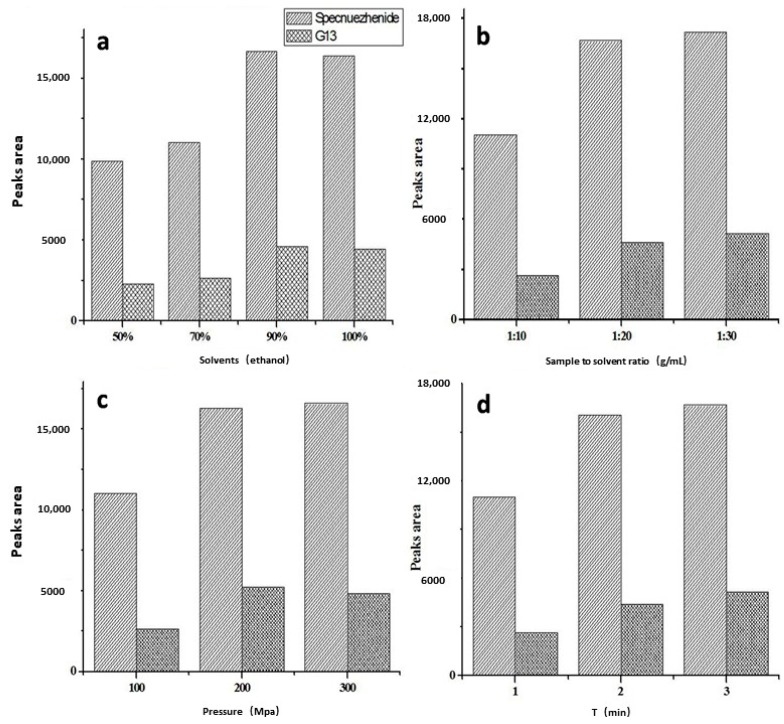
Effects of solvents (**a**), sample to solvent ratio (g/mL) (**b**), pressure (**c**) and extraction time (**d**) on the changes in nuezhenoside G13 (**4**) and specnuezhenide (**5**) contents by UPE.

**Figure 3 molecules-23-03353-f003:**
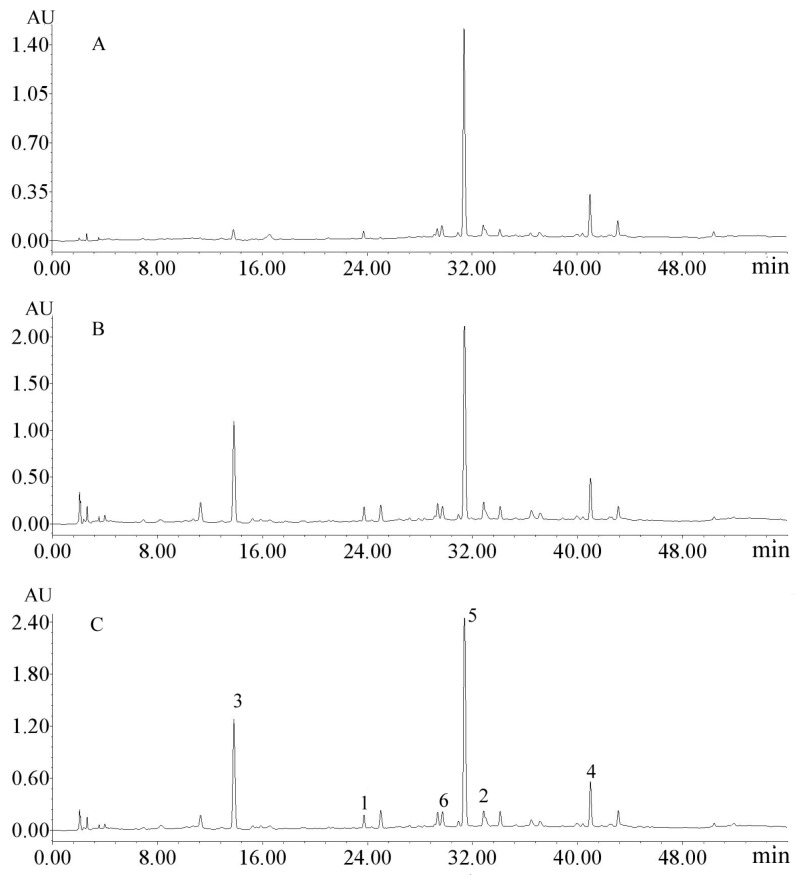
HPLC chromatograms of the extraction samples by ultrasonic (**A**), heat reflux extraction (**B**), and UPE (**C**) from Ligustri Lucidi Fructus. HPLC conditions: column, Purospher STAR RP18 Endiapped (250 mm × 4.6 mm i.d., 5 μm); mobile phase, acetonitrile (**A**)-water (**B**), 0–2 min, 5–5%A; 2–40 min, 5–30%A; 40–45 min, 30–30%A; 45–53 min, 30–50%A; column temperature, 25 °C; flow rate, 1.0 mL·min^−1^; UV detection wavelength, 282 nm; injection volume, 10 μL; peak 1, salidroside; peak 2, echinacoside; peak 3, acteoside; peak 4, specnuezhenide; peak 5, isonuezhenide; and peak 6, nuezhenoside G13.

**Figure 4 molecules-23-03353-f004:**
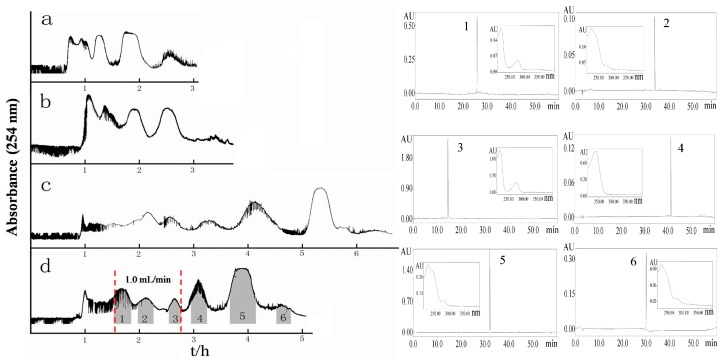
High-speed counter-current chromatography (HSCCC) chromatogram for the separation of Ligustri Lucidi Fructus UPE extract and HPLC analysis of the peak fractions. Experimental conditions: solvent system, ethyl acetate:*n*-butanol:water (2:1:3, *v*/*v*); the upper organic phase as the stationary phase and the lower aqueous phase as the mobile phase; revolution speed, 800 rpm; flow rate, 0–100 min, 2.0 mL·min^−1^; 100–170 min, 1.0 mL·min^−1^; 170–300 min, 2.0 mL·min^−1^; sample size, 200 mg; UV detection wavelength, 254 nm; retention of stationary phase, 65.7%.

**Table 1 molecules-23-03353-t001:** Partition coefficients (*K*_D_) of the target compounds in different solvent systems.

Solvent System Ethyl Acetate:*n*-Butanol:Water	Peak No.					
	1(*K*_D_)	2(*K*_D_)	3(*K*_D_)	4(*K*_D_)	5(*K*_D_)	6(*K*_D_)
1:0:1	0.11	0.10	0.13	0.05	0.23	0.06
4:1:5	0.56	0.49	1.39	0.65	0.46	0.34
2:1:3	0.58	0.50	1.38	0.78	0.55	0.46

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
