# Peer review of "Preparative Separation of Phenylethanoid and Secoiridoid Glycosides from Ligustri Lucidi Fructus by High-Speed Counter-Current Chromatography Coupled with Ultrahigh Pressure Extraction"

_molecules, 2018, doi:10.3390/molecules23123353_

Round 1
Reviewer 1 Report
The Authors described the separation on preparative scale of pPhenylethanoid and secoiridoid glycosides from Ligustri Lucidi fructus by High-Speed Counter-Current Chromatography coupled with Ultrahigh Pressure Extraction. The development of methods for receiving of biologically active compounds from plant extracts is still an important aspect of scientific research. The Authors applied relatively rapid and efficient extraction method for extraction of phenylethanoid and secoiridoid glycosides from Ligustri Lucidi fruits. They analyzed a numerous parameters influenced on extraction efficiency such as composition of solvents, sample to solvent ratio, pressure and extraction time. Based on the conducted optimization, they chose the most optimal conditions for extraction of these compounds.
In my opinion the Authors should further emphasize the advantages of the developed method and the novelty of the manuscript especially in conclusions.
Author Response
Response to Reviewer 1 Comments
Comments and Suggestions for Authors
The Authors described the separation on preparative scale of pPhenylethanoid and secoiridoid glycosides from Ligustri Lucidi fructus by High-Speed Counter-Current Chromatography coupled with Ultrahigh Pressure Extraction. The development of methods for receiving of biologically active compounds from plant extracts is still an important aspect of scientific research. The Authors applied relatively rapid and efficient extraction method for extraction of phenylethanoid and secoiridoid glycosides from Ligustri Lucidi fruits. They analyzed a numerous parameters influenced on extraction efficiency such as composition of solvents, sample to solvent ratio, pressure and extraction time. Based on the conducted optimization, they chose the most optimal conditions for extraction of these compounds.
In my opinion the Authors should further emphasize the advantages of the developed method and the novelty of the manuscript especially in conclusions.
Point 1: In my opinion the Authors should further emphasize the advantages of the developed method and the novelty of the manuscript especially in conclusions.
Response 1: We have emphasized the advantages of the developed method and the novelty of the manuscript in conclusions as ‘In this present work, a combined method for rapid extraction and efficient separation of three phenylethanoid glycosides, echinacoside (1), salidroside (3), acteoside (6), and three secoiridoid glycosides, isonuezhenide (2), nuezhenoside G13 (4), specnuezhenide (5), from Ligustri Lucidi Fructus, has been successfully achieved by UPE and HSCCC for the first time. The optimal UPE extraction was developed with conditions at solvent of 90% ethanol, sample/solvent ratio of 1:20 g/mL, pressure of 200 MPa, and time of 2 min, which rendered the yields of 4 and 5 to be 15.0 and 78.0 mg/g, respectively. These experimental results indicated that UPE extraction needed shorter time and less solvent, however, produced higher yields, when compared with conventional extraction methods. Meanwhile, the HSCCC separation strategy of changing flow rates between 1.0 and 2.0 mL/min allowed the acquisition of these six compounds in one separation run in about 300 min, which indicated that HSCCC could yield compounds of widely varied polarities with less time and solvent consuming compared with conventional column separation. The overall results showed that the combination of UPE and HSCCC is efficient for the extraction and isolation of phenylethanoid and secoiridoid glycosides from Ligustri Lucidi Fructus. To the best of our knowledge, this integration method of UPE and HSCCC was applied to rapidly obtain the active components from Ligustri Lucidi Fructus for the first time. What’s more, this gradient flow rate method gave good performance in isolating compounds with similar polarities and poor peak resolutions, which paved the pathway for one-run systematic separation of complex compounds.’.
Reviewer 2 Report
The article is clearly and simply written. There are just few minor points to correct
Line 288 replace "decocted" with "extracted".
One hour is rather a long time for ultrasonic extraction. Were there attempts to extract the sample for shorter time or less amount of solvent? If not, it should be clearly stated in results and discussion when mentioning the comparison among techniques.
Why were echinacoside, salidroside, acteoside, isonuezhenide, nuezhenoside G13 and specnuezhenide selected as target compounds? Please add a sentence or two in the introduction.
Please remove the highlight from supplementary material.
Author Response
Response to Reviewer 2 Comments
Comments and Suggestions for Authors
The article is clearly and simply written. There are just few minor points to correct
Point 1: Line 288 replace "decocted" with "extracted".
Response 1: In this revised manuscript, "decocted" in line 288 has been replaced by "extracted".
Point 2: One hour is rather a long time for ultrasonic extraction. Were there attempts to extract the sample for shorter time or less amount of solvent? If not, it should be clearly stated in results and discussion when mentioning the comparison among techniques.
Response 2: The extraction time for ultrasonic is usually 30 min. And here the time is determined to be one hour the same as that of heat extraction, which is used to compare the results of the crude extract obtained by the two different traditional treatment methods. What’s more, we have added these in 2.1 Ultrahigh pressure extraction parameters part when mentioning the comparison among techniques as ‘The extractabilities of phenylethanoid and secoiridoid glycosides from Ligustri Lucidi Fructus carried out by UPE and traditional heat reflux and ultrasonic experiments were compared. The traditional extractions were carried out with the same extraction time of 1 h and sample/solvent ratio of 1:20 g/mL. The analysis results of the crude extract obtained by HPLC chromatograms of the three different treatment methods characterized similarly, except for the more extraction yields of compounds 3, 4, and 5 by UPE than the other two extractions in Fig. 3. Moreover, to achieve the same amounts of purified compounds 3, 4, and 5, the extraction time and solvent amount of the two traditional extraction methods are much more than those of UPE. Thus, the UPE pretreatment was confirmed to be efficiency for extraction phenylethanoid and secoiridoid glycosides from Ligustri Lucidi Fructus.’.
Point 3: Why were echinacoside, salidroside, acteoside, isonuezhenide, nuezhenoside G13 and specnuezhenide selected as target compounds? Please add a sentence or two in the introduction.
Response 3: The reason why these six compounds being selected as target compounds has been added in the 1. Introduction part as ‘A literature survey indicated that phenylethanoid glycosides in Ligustri Lucidi Fructus were mainly composed of salidroside, echinacoside, acteoside, and cimidahurinine, as well as secoiridoid glycosides mainly composed of specnuezhenide, isonuezhenide, and nuezhenoside G13 [7]. Additionally, salidroside and specnuezhenide are representative compounds of Ligustri Lucidi Fructus, with the latter used as the quality control marker in the Chinese Pharmacopoeia 2015 [10].’. And one more reference [10] Pharmacopoeia Commission of the Ministry of Public Health. 2015. Chinese pharmacopoeia, Part I. Beijing: China Medical Science and Technology Press, 45-46. has been added in this revised manuscript.
Point 4: Please remove the highlight from supplementary material.
Response 4: In this revised manuscript, the highlight from supplementary material has been removed.